# Hunger- and thirst-sensing neurons modulate a neuroendocrine network to coordinate sugar and water ingestion

**Amanda J González Segarra\*, Gina Pontes[†], Nicholas Jourjine[‡], Alexander Del Toro[§], Kristin Scott\***

University of California, Berkeley, Berkeley, United States

**Abstract** Consumption of food and water is tightly regulated by the nervous system to maintain internal nutrient homeostasis. Although generally considered independently, interactions between hunger and thirst drives are important to coordinate competing needs. In *Drosophila*, four neurons called the interoceptive subesophageal zone neurons (ISNs) respond to intrinsic hunger and thirst signals to oppositely regulate sucrose and water ingestion. Here, we investigate the neural circuit downstream of the ISNs to examine how ingestion is regulated based on internal needs. Utilizing the recently available fly brain connectome, we find that the ISNs synapse with a novel cell-type bilateral T-shaped neuron (BiT) that projects to neuroendocrine centers. In vivo neural manipulations revealed that BiT oppositely regulates sugar and water ingestion. Neuroendocrine cells downstream of ISNs include several peptide-releasing and peptide-sensing neurons, including insulin producing cells (IPCs), crustacean cardioactive peptide (CCAP) neurons, and CCHamide-2 receptor isoform RA (CCHa2R-RA) neurons. These neurons contribute differentially to ingestion of sugar and water, with IPCs and CCAP neurons oppositely regulating sugar and water ingestion, and CCHa2R-RA neurons modulating only water ingestion. Thus, the decision to consume sugar or water occurs via regulation of a broad peptidergic network that integrates internal signals of nutritional state to generate nutrient-specific ingestion.

**\*For correspondence:**
amandagonzalez@berkeley.edu
(AJG-S);
kscott@berkeley.edu (KS)

**Present address:** [†]IBBEA, CONICET-UBA, Buenos Aires, Argentina; [‡]Harvard University, Cambridge, United States; [§]Brown University, Rhode Island, United States

**Competing interest:** The authors declare that no competing interests exist.

## eLife assessment

This **important** study identifies and characterizes a broad peptidergic network that coordinates nutrient-specific consumption needs for food or water. Using state-of-the-art methodology the authors combine a well-balanced set of exploratory anatomical analyses with rigorous functional experimental approaches to examine how ingestion is regulated based on internal needs. These significant and **convincing** new findings are of broad interest to the neuroscience field.

## Introduction

The survival of an organism depends on its ability to coordinate nutrient ingestion with internal nutrient abundance in order to meet its metabolic needs. The nervous system acts as an internal nutrient abundance sensor to drive ingestion in nutrient-deprived states and inhibit ingestion in nutrient-replete states to restore homeostasis (*Gizowski and Bourque, 2018*; *Jourjine, 2017*; *Sternson et al., 2013*; *Qi et al., 2021*; *Yoshinari et al., 2021*). Although generally considered independently, recent studies have demonstrated that interactions between hunger and thirst signals coordinate competing needs (*Burnett et al., 2016*; *Cannell et al., 2016*; *Jourjine et al., 2016*; *Watts and Boyle, 2010*; *Zimmerman et al., 2016*).

In mammals, regulation of hunger and thirst drives likely occurs through interactions between food and water ingestion circuits (*Eiselt et al., 2021*). In the arcuate nucleus of the hypothalamus, neurons that express the agouti related peptide (AgRP) and neuropeptide Y promote food ingestion while neurons that express pro-opiomelanocortin inhibit food ingestion (*Aponte et al., 2011*; *Graham et al., 1997*; *Sternson et al., 2013*). These neurons can detect circulating ghrelin, glucose, insulin, and leptin secreted from peripheral organs, in addition to receiving input from the gut through the vagus nerve (*Sternson et al., 2013*). In the subfornical organ, neurons expressing neuronal nitric oxide synthase (nNOS) promote water ingestion while neurons expressing the vesicular GABA transporter inhibit water ingestion. These cells directly detect blood osmolality and receive input from the gut via the vagus nerve and from the mouth via the trigeminal nerve (*Gizowski and Bourque, 2018*; *Zhang et al., 2022a*). Interestingly, activation of AgRP neurons decreases water ingestion and inhibition of nNOS expressing cells increases food ingestion (*Burnett et al., 2016*; *Zimmerman et al., 2016*). This suggests that hunger-sensing cells promote food ingestion and inhibit water ingestion, while thirst-sensing cells do the opposite (*Jourjine, 2017*). However, the underlying circuit mechanisms that lead to this reciprocal coordination of hunger and thirst remain unexplored.

Because of its numerically less complex nervous system, complete connectome, and abundant genetic tools, *Drosophila* is an ideal organism in which to study the coordination of hunger and thirst (*Pfeiffer et al., 2008*). Like mammals, *Drosophila melanogaster* selectively consumes food when hungry and water when thirsty (*Dethier, 1976*; *Gáliková et al., 2018*; *Landayan et al., 2021*; *Lin et al., 2014*; *Min et al., 2016*; *Yapici et al., 2016*). Moreover, in *Drosophila*, two pairs of neurons, the interoceptive subesophageal zone neurons (ISNs), directly integrate hunger and thirst signals to oppositely regulate sugar and water ingestion (*Jourjine et al., 2016*).

The ISNs express the adipokinetic hormone receptor, a G-protein coupled receptor which binds to the glucagon-like peptide adipokinetic hormone (AKH), a hormone released from the corpora cardiaca during starvation that signals nutrient deprivation (*Orchard, 1987*; *Gáliková et al., 2015*). AKH increases ISN activity to drive sugar ingestion and reduce water ingestion. The ISNs also express the TRPV channel Nanchung, which senses changes in hemolymph osmolality. High hemolymph osmolality, such as that experienced during thirst, decreases ISN activity to promote water ingestion and inhibit sugar ingestion (*Jourjine et al., 2016*). Thus, the ISNs sense both AKH and hemolymph osmolality, arguing that they balance internal osmolality fluctuations and nutrient need (*Jourjine et al., 2016*). How the ISNs achieve these effects on ingestion remains unclear.

To investigate how the ISNs transform internal nutrient detection into changes in feeding behaviors, we examined the neural network downstream of the ISNs. Using the fly brain connectome, intersectional genetic approaches, in vivo functional imaging, and behavioral assays, we identified a neural circuit downstream of the ISNs that regulates sugar and water ingestion. Our work reveals that the ISNs communicate with the neuroendocrine center of the fly brain and regulate the activity of a large number of neurons that transmit or receive peptidergic signals of nutritive state to bidirectionally regulate sugar and water ingestion.

## Results
### The ISNs are peptidergic neurons that release dILP3

To examine how the ISNs reciprocally regulate sugar and water ingestion, we aimed to identify the neural circuit downstream of the ISNs. We first sought to identify which neurotransmitter the ISNs use to communicate with downstream neurons. We expressed RNAi against enzymes involved in neurotransmitter synthesis, vesicular transporters, and neuropeptides in the ISNs and monitored water ingestion in water-deprived flies (*Figure 1A*). As decreasing activity of the ISNs increases water ingestion (*Jourjine et al., 2016*), we anticipated that an RNAi against the ISN neurotransmitter would decrease neurotransmission and increase water ingestion. Interestingly, in an RNAi screen of 18 common neurotransmitters and neuropeptides, only suppression of *Drosophila* insulin-like peptide 3 (dILP3) in the ISNs altered water ingestion (*Figure 1A*).

To confirm that dILP3 functions in the ISNs and to test whether it is involved in the reciprocal regulation of water and sugar ingestion, we expressed RNAi against dILP3 in the ISNs and measured sugar or water ingestion in water sated or thirsty flies, respectively (*Figure 1B*). As an additional approach to reduce dILP3, we expressed an RNAi against a neuropeptide processing protease, *amontillado*

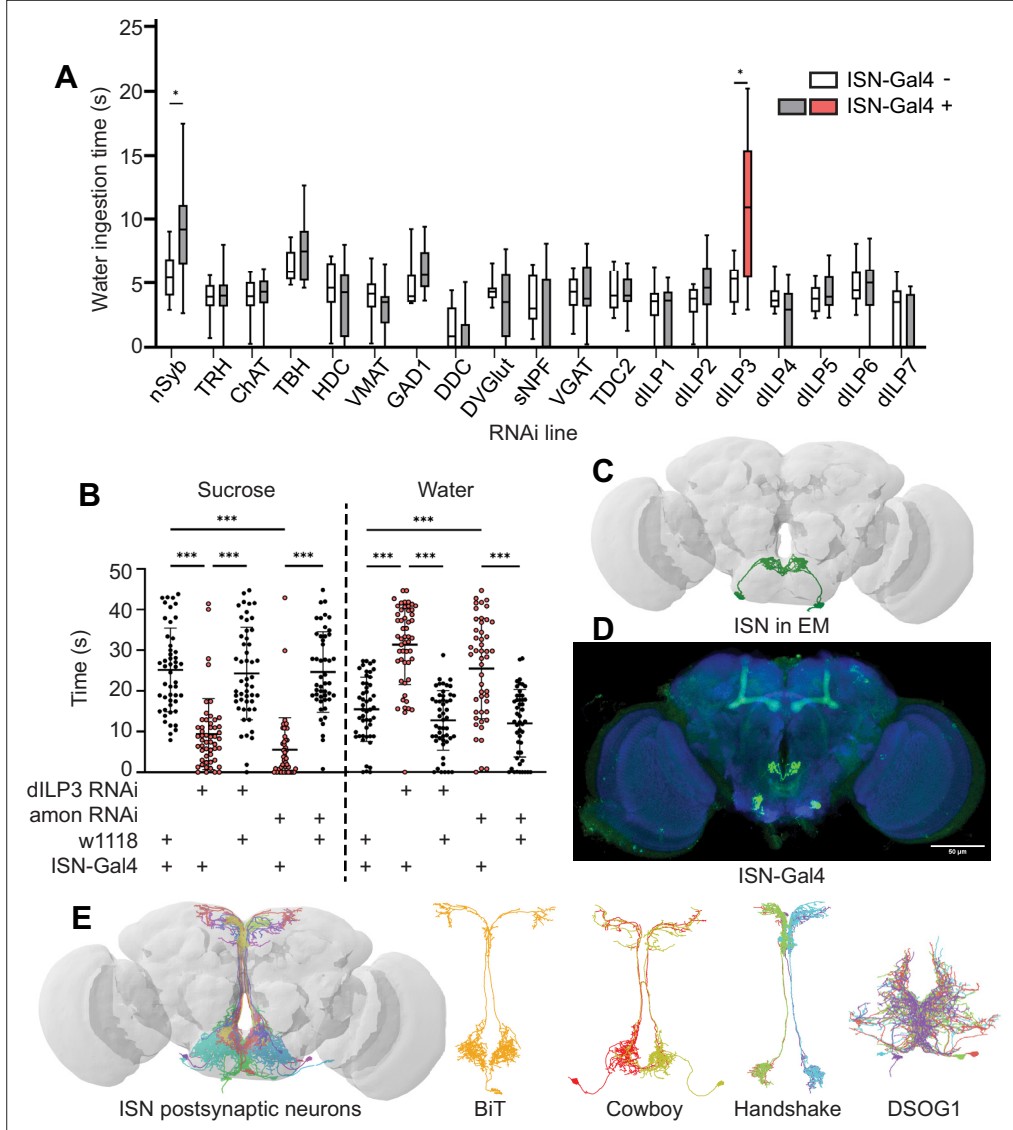

**Figure 1.** Interoceptive subesophageal zone neurons (ISNs) relay information to the pars intercerebralis. (**A**) Temporal consumption assay screen for water ingestion using RNAi targeting different neurotransmitter pathways. UAS-RNAi+ or - ISN-Gal4. RNAi against: nSynaptobrevin (nSyb), tryptophan hydroxylase (TRH), choline acetyltransferase (ChAT), tyrosine beta-hydroxylase (TBH), histamine decarboxylase (HDC), vesicular monoamine transporter (VMAT), glutamic acid decarboxylase 1 (GAD1), dopa decarboxylase (DDC), *Drosophila* vesicular glutamate transporter (DVGlut), short neuropeptide F (sNPF), vesicular GABA transporter (VGAT), tyrosine decarboxylase 2 (TDC2), *Drosophila* insulin-like peptide 1 (dILP1), *Drosophila* insulin-like peptide 2 (dILP2), *Drosophila* insulin-like peptide 3 (dILP3), *Drosophila* insulin-like peptide 4 (dILP4), *Drosophila* insulin-like peptide 5 (dILP5), *Drosophila* insulin-like peptide 6 (dILP6), *Drosophila* insulin-like peptide 7 (dILP7). Represented are the mean, and the 10–90 percentile; data was analyzed using Kruskal-Wallis test, followed by multiple comparisons against the RNAi control; p-values were adjusted using false discovery rate. n=8–39 animals/genotype except nSyb positive control (70–72). (**B**) Temporal consumption assay for 1 M sucrose or water using RNAi targeting dILP3 or amontillado in ISNs. Sucrose assay: Kruskal-Wallis test followed by Dunn's multiple comparison tests against ISN control and respective RNAi control. Water assay: ANOVA, Šídák's multiple comparison test to ISN control and respective RNAi control. n=48–52 animals/genotype. (**C**) ISNs reconstruction from full adult fly brain (FAFB) volume. (**D**) Light microscopy image of ISN-Gal4 registered to JFRC2010. (**E**) ISN postsynaptic neurons based on synapse predictions using FAFB volume (***Zheng et al., 2018***) and connectome annotation versioning engine (CAVE, ***Buhmann et al., 2021***; ***Ida et al., 2012***). Left: 10 postsynaptic neurons, right: postsynaptic neurons bilateral T-shaped neuron (BiT), Cowboy, Handshake, and DSOG1. *p<0.05, ***p<0.001.

The online version of this article includes the following source data and figure supplement(s) for figure 1:

*Figure 1 continued on next page*

*Figure 1 continued*

**Source data 1.** ISN neurotransmitter screen.

**Figure supplement 1.** Interoceptive subesophageal zone neuron (ISN) postsynaptic partners labeled by trans-Tango and EM.

(*Siekhaus and Fuller, 1999*), in the ISNs and tested sugar and water ingestion. We found that knockdown of either dILP3 or *amontillado* in the ISNs caused both a decrease in sugar ingestion and an increase in water ingestion (*Figure 1B*). This is the same phenotype that was previously reported in the ISNs upon loss of neurotransmission (*Jourjine et al., 2016*). These data argue that the ISNs are peptidergic neurons that release dILP3 and that one function of dILP3 is to promote sugar ingestion and inhibit water ingestion.

## The ISNs synapse onto neurons that arborize in neuroendocrine and feeding centers

*Drosophila* has one insulin-like receptor (dInR), a tyrosine kinase type receptor homologous to the human insulin receptor, which binds dILP3 and six of the additional *Drosophila* insulin-like peptides (*Brogiolo et al., 2001*; *Claeys et al., 2002*; *Clancy et al., 2001*; *Fernandez et al., 1995*; *Grönke et al., 2010*; *Nässel and Vanden Broeck, 2016*; *Tatar et al., 2001*). In adult flies, insulin signaling has been shown to regulate an array of physiological processes including metabolism, feeding, reproduction, and lifespan (*Badisco et al., 2013*; *Biglou et al., 2021*; *Clancy et al., 2001*; *Nässel et al., 2013*; *Ohhara et al., 2018*). Since dInR is ubiquitous and involved in many different processes (*Chen et al., 1996*; *Garofalo, 2002*; *Veenstra et al., 2008*), we could not leverage neurotransmitter receptor identity for postsynaptic neuron identification. We instead used the *trans*-Tango system (*Talay et al., 2017*), a genetic trans-synaptic tracer, to label neurons postsynaptic to the ISNs (*Figure 1—figure supplement 1A*). We expressed the *trans*-Tango ligand in the ISNs and its receptor panneurally. Binding of the ligand to its receptor induces GFP expression in the receptor-expressing cells and labels potential synaptic partners (*Talay et al., 2017*). *trans*-Tango labeling revealed numerous ISN postsynaptic arborizations in the subesophageal zone (SEZ), a brain region associated with taste processing and feeding circuits (*Gordon and Scott, 2009*; *Scott et al., 2001*; *Wang et al., 2004*), and along the median bundle to the superior medial protocerebrum (SMP), a neuroendocrine center (*Hartenstein, 2006*; *Nässel and Zandawala, 2020*; *Figure 1—figure supplement 1A*). However, as many ISN candidate postsynaptic neurons were labeled, the morphology of individual neurons was unclear.

To comprehensively examine the postsynaptic partners of the ISNs, we employed the full adult fly brain (FAFB) volume, a whole-brain electron microscopy volume that provides synaptic resolution of all neurons in the fly brain (*Zheng et al., 2018*). We manually reconstructed the ISNs using CATMAID (*Li et al., 2019*; *Saalfeld et al., 2009*) by tracing neuronal arbors from the pharyngeal nerve with large cell bodies in the SEZ. Due to the ISNs' unique morphology, with large cell bodies near the pharyngeal nerve and dense neurites in the flange that cross the midline, we used visual morphological comparison of the reconstructed ISNs in the FAFB volume (*Figure 1C*) and light microscopy images of *ISN-Gal4* (*Figure 1D*) to identify the ISNs. Once we had reconstructed the ISNs, we labeled presynaptic sites in the ISNs and postsynaptic sites in other neurons based on known synapse active zone structure (*Zhai and Bellen, 2004*). We then reconstructed neurons that were postsynaptic to the ISNs.

Soon after we had reconstructed the four ISNs and several postsynaptic neurons in CATMAID, the FlyWire whole-brain connectome of more than 80,000 reconstructed EM neurons became available (*Dorkenwald et al., 2022*, flywire.ai). Since FlyWire uses the FAFB volume, we used the coordinates of the ISNs we traced in CATMAID to locate them in FlyWire. Additionally, we compared a pointcloud generated from a registered light microscopy image of *ISN-Gal4* (*Figure 1D*) to the reconstructed ISNs in the FAFB volume (*Figure 1C*) to further confirm ISN identity. We identified neurons downstream of the ISNs (*Figure 1E*). We found that the ISNs have 104 predicted postsynaptic partners with five or more synapses, comprising nine morphological cell types (*Supplementary file 1*, *Figure 1—figure supplement 1B–K*). These include known cell types (Cowboy, DSOG1, FLAa2, FLAa3/Lgr3, and the ISNs; *Lee et al., 2020*; *Pool et al., 2014*; *Sterne et al., 2021*; *Yu et al., 2013*) as well as many uncharacterized cell types. The ISN predicted postsynaptic partners include projection neurons that project along the median bundle to the SMP (64 cells), local SEZ neurons (18 cells), ascending neurons

with projections coming through the neck connective (10 cells), descending neurons with projections leaving through the neck connective (8 cells), and the ISNs themselves (4 cells). This connectivity is consistent with the connectivity determined by *trans*-Tango (*Figure 1—figure supplement 1A*). Overall, the ISN synaptic connectivity suggests that the hunger and thirst signals sensed by the ISNs are conveyed to a broad network, with the potential to coordinate feeding behaviors (SEZ neurons), nutrient status (SMP neuroendocrine centers), and movement or digestion (ascending and descending neurons). We note that as neuropeptides may diffuse over long distances (*van den Pol, 2012*), ISN dILP3 release may also influence activity of additional neurons that are not synaptically connected to the ISNs.

## The ISN postsynaptic neuron BiT reciprocally regulates sugar and water ingestion

As the majority of the ISN predicted postsynaptic partners project to the SMP, we examined whether ISN communication to this region regulates neuroendocrine cells and/or influences feeding behavior. As a first step, we focused on an uncharacterized neuron that receives the most synaptic input from the ISN per single cell. We named this neuron bilateral T-shaped neuron (BiT). BiT has its cell body in the SEZ and bilateral projections in the flange and SMP. It receives 7.4% of ISN synaptic output (301/4050 synapses) (*Figure 1—figure supplement 1B* and *Supplementary file 1*). In turn, the ISNs are the main synaptic input to BiT, comprising 17% of BiT's synaptic input (301/1763 synapses). We generated a split-Gal4 line that labels BiT to study its function (*Figure 2B*). We screened over 20 AD-DBD combinations and found that *VT002073-Gal4.AD* and *VT040568-Gal4.DBD* specifically labeled BiT. We confirmed this by comparing a pointcloud generated from a registered light microscopy image of *BiT split-Gal4* (*Figure 2B*) with the reconstructed BiT in the FAFB volume (*Figure 2A*).

To test whether the ISNs are functionally connected to BiT, we conducted in vivo functional imaging experiments in which we activated the ISNs while simultaneously monitoring BiT's neural activity. We expressed the light activated cation channel Chrimson in the ISNs and the voltage sensor ArcLight in BiT (*Figure 2C*; *Jin et al., 2012*; *Klapoetke et al., 2014*). In one experiment, we applied two consecutive 2 s stimulations (*Figure 2D*) to test whether the response was reproducible. In another experiment, we applied a longer 30 s stimulation (*Figure 2E*) to ensure we captured the full response to ISN stimulation since dILPs can act over longer time scales (*Sudhakar et al., 2020*). In both experiments, we found that stimulating the ISNs increased ArcLight fluorescence in BiT, demonstrating that BiT became hyperpolarized (*Figure 2D, E*, *Figure 2—figure supplement 1*). Oscillation in BiT's response during the 30 s stimulation (*Figure 2E*) is due to oscillations in the LED stimulation paradigm. Thus, increased activity in the ISNs inhibits BiT.

Next, we tested whether BiT modulates sugar or water ingestion. We measured total ingestion time of sugar or water while activating or inhibiting BiT. We found that acute optogenetic activation of BiT decreased sugar ingestion and increased water ingestion (*Figure 2F*, *Figure 2—figure supplement 2*). Moreover, reducing synaptic transmission in BiT using nSynaptobrevin (nSyb) RNAi caused increased sugar ingestion and decreased water ingestion (*Figure 2G*, *Figure 2—figure supplement 2*). These data demonstrate that BiT is both necessary and sufficient to regulate sugar and water ingestion. Furthermore, we find that the activation and silencing phenotypes for BiT are opposite to the ISN phenotypes, consistent with our calcium imaging studies that the ISNs inhibit BiT. These findings reveal that the coordination of sugar and water ingestion is maintained downstream of the ISNs.

These studies demonstrate that BiT activity reciprocally regulates sugar and water ingestion, similar to the ISNs. Hunger signals (i.e. AKH) activate the ISNs, causing the ISNs to inhibit BiT, which in turn increases sugar ingestion. On the other hand, thirst signals (i.e. high hemolymph osmolality) inhibit the ISNs, releasing ISN inhibition onto BiT, causing an increase in water ingestion (*Figure 2H*). Strikingly, although BiT is only one ISN downstream neuron, its activity increases and decreases are sufficient to coordinate both sugar and water ingestion, suggesting that it is a critical node in the ISN network.

## BiT downstream partners include neuroendocrine cells that convey nutritional status

To examine how BiT coordinates sugar and water ingestion, we investigated the neural circuit downstream of BiT using the FlyWire connectome (*Figure 3*). The FAFB connectivity revealed that BiT has 93 predicted postsynaptic partners. Unlike the ISNs' downstream partners, which only innervate the

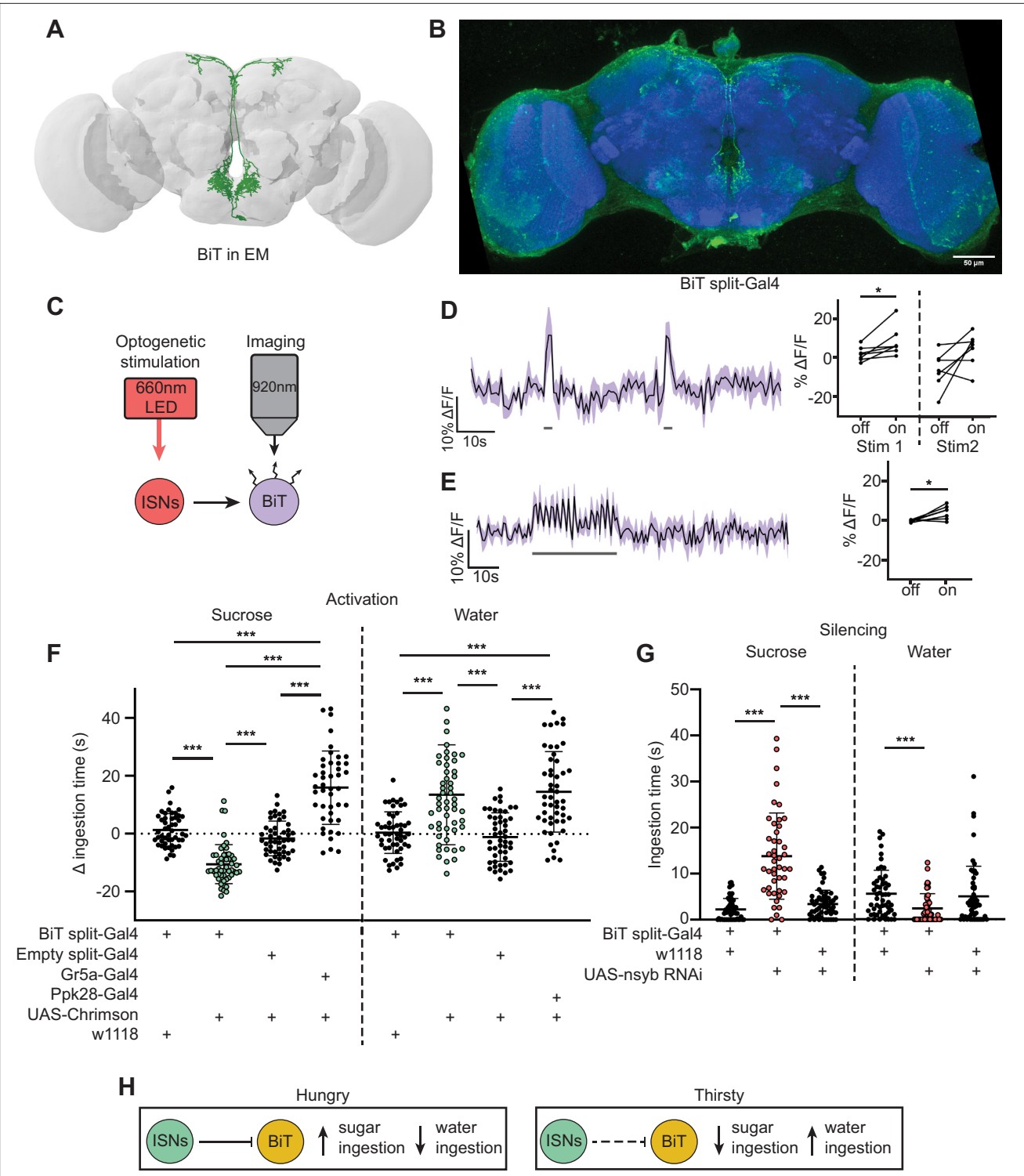

**Figure 2.** Interoceptive subesophageal zone neurons (ISNs) inhibit bilateral T-shaped neuron (BiT), which oppositely regulates sugar and water ingestion. (**A**) BiT neuron reconstruction from full adult fly brain (FAFB) dataset. (**B**) Light microscopy image of BiT split-Gal4. (**C**) Experimental setup for in vivo voltage imaging. We expressed the light-sensitive ion channel Chrimson in the ISNs and optogenetically stimulated them with 660 nm LED. We expressed the voltage sensor ArcLight in BiT and imaged it with a two-photon microscope. (**D**) ArcLight response of BiT soma to 2 s optogenetic stimulation of the ISNs or (**E**) 30 s optogenetic stimulation of the ISNs. Left: Scatter plot shows mean ± SEM of all flies imaged, gray bars represent LED stimulation. Right: Quantification of mean fluorescence intensity before stim (off) and during stim (on), each dot represents one fly. Paired Wilcoxon and paired t-test (Stim 2, p=0.07). n=7 flies. (**F**) Temporal consumption assay for 1 M sucrose or water during acute optogenetic activation of BiT with

*Figure 2 continued on next page*

*Figure 2 continued*

Chrimson. Ingestion time of females exposed to light normalized to dark controls of indicated genotype. Sucrose: Kruskal-Wallis test with Dunn's multiple comparison test. Water: One-way ANOVA with Holm-Šídák multiple comparison test. n=44–54 animals/genotype. (**G**) Temporal consumption assay for 1 M sucrose or water using RNAi targeting nSyb in BiT. Kruskal-Wallis with Dunn's multiple comparison test. n=45–57 animals/genotype. (**H**) Neural model for BiT coordination of sucrose and water intake. Dashed lines indicate inactive synapses. *p<0.05, ***p<0.001.

The online version of this article includes the following source data and figure supplement(s) for figure 2:

**Source data 1.** BiT functional imaging.

**Figure supplement 1.** Bilateral T-shaped neuron (BiT) genetic control functional imaging.

**Figure supplement 2.** Bilateral T-shaped neuron (BiT) optogenetic activation ingestion phenotype.

SMP and SEZ, BiT postsynaptic partners reach more brain regions including the superior lateral protocerebrum (SLP), fan shaped body (FB), lobula, SMP, and SEZ. This suggests that the hunger and thirst signals detected by the ISNs are conveyed by BiT to widely regulate brain activity.

Many of the BiT predicted postsynaptic partners arborize in both the SEZ and SMP, suggesting that they might coordinate nutritional status and feeding. Several BiT targets transmit or receive peptidergic signals of nutrient state. For example, BiT postsynaptic partners include insulin producing cells (IPCs), FLAa3/Lgr3 neurons, and neurons labeled by the *CCHa2R-RA-Gal4* line (***Deng et al., 2019***; *Figure 3*, *Supplementary file 2*). IPCs are a well-studied cell type that release dILP2, dILP3, and dILP5, regulate glucose uptake, and influence many physiological processes including feeding (***Nässel et al., 2013***; ***Ohhara et al., 2018***). FLAa3/Lgr3 neurons detect dILP8 and influence sugar ingestion (***Meissner et al., 2016***; ***Yeom et al., 2021***; ***Yu et al., 2013***). CCHa2 and its receptor CCHa2R have been shown to participate in feeding regulation and regulate insulin signaling, although the function of CCHa2R-RA neurons has not been examined (***Deng et al., 2019***; ***Ida et al., 2012***; ***Ren et al., 2015***; ***Sano et al., 2015***; ***Shahid et al., 2021***). Thus, BiT is predicted to synapse onto many neuroendocrine neurons, possibly enabling integration of the hunger and thirst signals sensed by ISNs with diverse nutrient state signals.

## IPCs regulate sugar and water ingestion

The IPCs integrate multiple signals of nutrient status and regulate feeding and metabolism (***Nässel and Zandawala, 2020***). We found that the ISNs are connected to the IPCs via BiT. BiT is the main synaptic input into IPCs, making up 25% of the IPCs' synaptic input (442/1735) and IPCs receive 25% of BiT's synaptic output (442/1742) (*Figure 3*, *Supplementary file 2*). We tested whether BiT is functionally connected to the IPCs by optogenetically stimulating BiT and monitoring activity in IPCs using the calcium sensor GCaMP6s (***Chen et al., 2013***). We found that BiT inhibits IPCs (*Figure 3—figure supplement 1A–D*), consistent with neurotransmitter predictions that BiT uses glutamate (***Eckstein et al., 2020***), which can act as an inhibitory neurotransmitter in *Drosophila* (***Liu and Wilson, 2013***).

To test whether IPCs modulate ingestion of sucrose or water under conditions that reveal ISN behavioral phenotypes, we measured ingestion time of sucrose or water while acutely activating the IPCs. We found that acute activation of IPCs increased sucrose ingestion and decreased water ingestion (*Figure 3—figure supplement 1E*, *Figure 3—figure supplement 2*). These results are consistent with one study (***Sudhakar et al., 2020***) but differ from other studies showing that acute IPC activation limits ingestion of sucrose or food (***Nässel et al., 2015***; ***Wang et al., 2020***). IPCs integrate many signals and release multiple peptides (***Sano et al., 2015***; ***Söderberg et al., 2012***; ***Ohhara et al., 2018***; ***Wang et al., 2020***), suggesting that differences in these behavioral results may, in part, stem from differences in the current nutritional state sensed by the IPCs. While further experiments are needed to elucidate how IPCs coordinate nutrient state and ingestion under different conditions, our results show that BiT regulates IPC activity and that IPC activity coordinates both sugar and water ingestion.

## CCHa2R-RA neurons regulate water ingestion downstream of BiT

A number of studies indicate that CCHa2 and its receptor CCHa2R promote food intake and appetite in various insects, including blowflies (***Ida et al., 2012***), aphids (***Shahid et al., 2021***), and *Drosophila* (***Ren et al., 2015***). BiT synapses with CCHa2R-RA neurons, four neurons with cell bodies in the SEZ, and arbors in the flange and pars intercerebralis (*Figure 4A, B*). BiT is the dominant input onto CCHa2R-RA

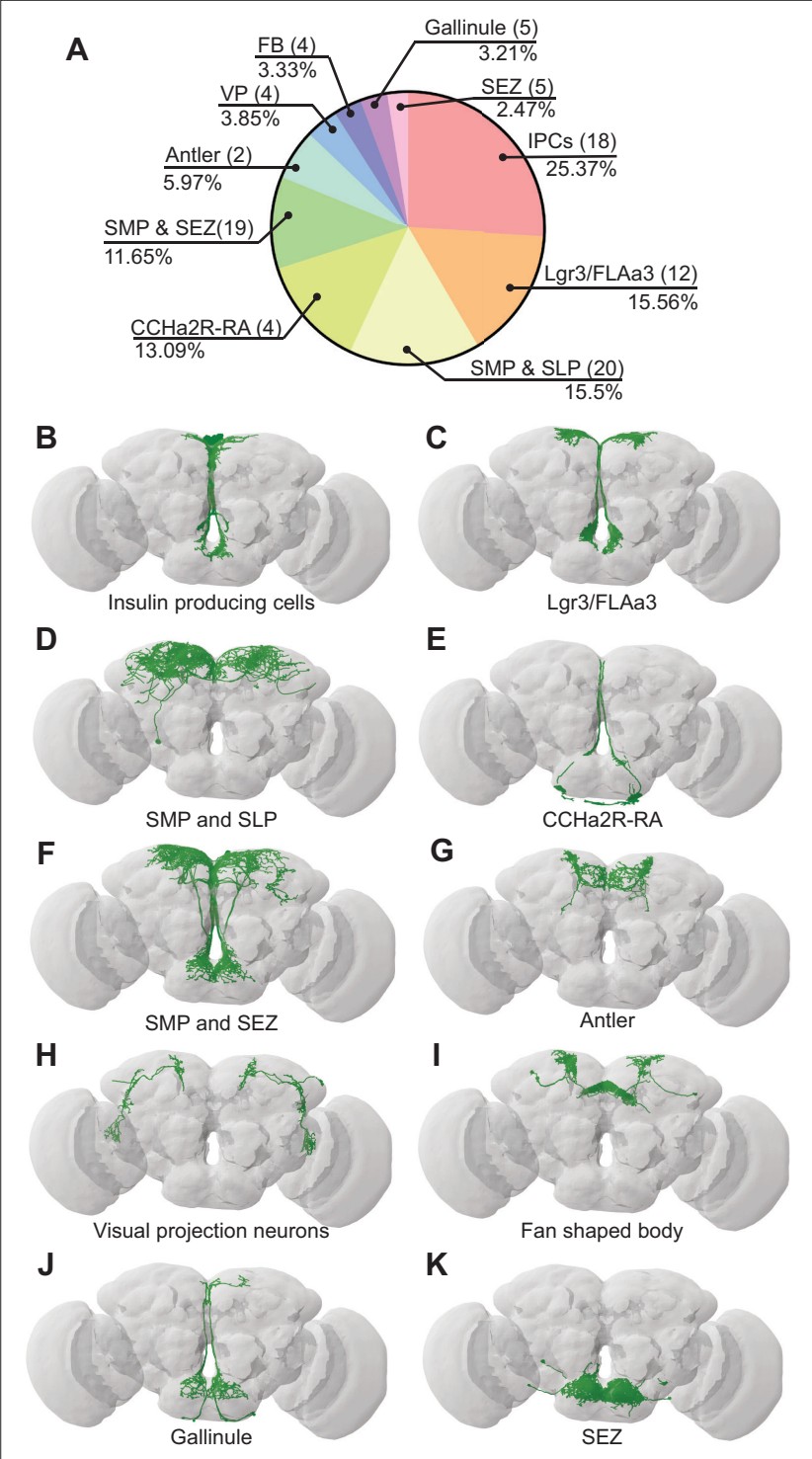

**Figure 3.** Bilateral T-shaped neuron (BiT) postsynaptic neurons include neuroendocrine cells. (**A**) Distribution of synaptic output from BiT divided by cell class or brain region. Total of 1742 synapses from BiT and 93 postsynaptic partners. Insulin producing cells (IPCs) (18 neurons) receive 25.37% of all BiT output, Lgr3/FLAa3 (12 neurons) 15.56%, superior medial protocerebrum (SMP) and superior lateral protocerebrum (SLP) (20 neurons) 15.5%, CCHamide-2 receptor isoform RA (CCHa2R-RA) (4 neurons) 13.09%, SMP and subesophageal zone (SEZ) (19 neurons) 11.65%, Antler (2 neurons) 5.97%, visual projections (4 neurons) 3.85%, fan shaped body (4 neurons) 3.33%, Gallinule (5 neurons) 3.21%, SEZ (5 neurons) 2.47%. Only postsynaptic partners with five or more synapses were considered for this analysis. Reconstruction of IPCs (**B**), Lgr3/FLAa3 neurons (**C**), neurons innervating the

*Figure 3 continued on next page*

*Figure 3 continued*

SMP and SLP (**D**), CCHa2R-RA neurons (**E**), neurons innervating the SMP and SEZ (**F**), Antler neurons (**G**), visual projection neurons (**H**), neurons innervating the fan shaped body (**I**), Gallinule neurons (**J**), and neurons innervating the SEZ (**K**).

The online version of this article includes the following source data and figure supplement(s) for figure 3:

**Source data 1.** IPCs functional imaging.

**Figure supplement 1.** Insulin producing cell (IPC) response to bilateral T-shaped neuron (BiT) stimulation.

**Figure supplement 2.** Insulin producing cell (IPC) optogenetic activation ingestion phenotype.

neurons, comprising 94% of CCHa2R-RA presynaptic sites (171/181 synapses). CCHa2R-RA neurons receive the most output from BiT per single cell comprising 13% of BiT's output (228/1742 synapses). To investigate whether BiT's synaptic input to CCHa2R-RA neurons regulates ingestion, we examined the functional connectivity between BiT and CCHa2R-RA neurons and the behavioral phenotypes associated with CCHa2R-RA neurons.

We monitored activity in CCHa2R-RA neurons with the calcium indicator GCaMP6s upon optogenetic stimulation of BiT; however, we did not observe a response in CCHa2R-RA neurons (*Figure 4— figure supplement 1D–E*). As BiT likely inhibits CCHa2R-RA neurons, it is possible that we were unable to detect an inhibitory response in CCHa2R-RA neurons using a calcium sensor. We therefore monitored activity of CCHa2R-RA neurons upon optogenetic stimulation of the ISNs, as the ISNs should activate CCHa2R-RA neurons given that the ISNs inhibit BiT (*Figure 4C*). Indeed, we found that CCHa2R-RA neurons showed robust calcium responses upon ISN stimulation relative to controls (*Figure 4D–E*, *Figure 4—figure supplement 1A–C*), demonstrating that these neurons are functionally connected to the ISNs, likely via BiT inhibition.

To test if CCHa2R-RA neurons regulate sugar or water ingestion, we manipulated activity in these neurons and measured ingestion of sugar or water. We found that activation of CCHa2R-RA neurons decreased water ingestion but did not change sugar ingestion (*Figure 4F*, *Figure 4—figure supplement 2*). Moreover, inhibiting neurotransmission in CCHa2R-RA neurons increased water ingestion (*Figure 4G*, *Figure 4—figure supplement 2*), but did not change sucrose ingestion relative to *CCHa2R-RA-Gal4* controls. These behavioral experiments demonstrate that peptide-sensing neurons downstream of the ISNs regulate water ingestion (*Figure 4H*). The finding that CCHa2-RA neurons recapitulate the water ingestion phenotypes of the ISNs but not sugar ingestion phenotypes suggests that the ISNs activate different arrays of peptidergic neurons that contribute differentially to ingestion of specific nutrients.

## CCAP neurons are downstream of the ISNs and reciprocally regulate sugar and water ingestion

In a separate effort to find neurons that are postsynaptic to the ISNs, we tested whether neurons that had previously been implicated in ingestion were functionally connected to the ISNs. We conducted pilot in vivo functional imaging experiments monitoring the activity of candidate neurons with GCaMP7b while optogenetically stimulating the ISNs. We found one set of peptidergic neurons, the crustacean cardioactive peptide (CCAP) neurons, that were activated upon ISN optogenetic stimulation (*Figure 5A–E*, *Figure 5—figure supplement 1*).

CCAP neurons have been shown to regulate feeding behavior in adult *Drosophila* as loss of CCAP in these neurons reduced sucrose ingestion (*Williams et al., 2020*). To directly test if CCAP neural activity modulates sugar or water ingestion, we acutely manipulated the activity of CCAP neurons and measured ingestion of sugar or water. We found that activation of CCAP decreased water ingestion and increased sugar ingestion (*Figure 5F*, *Figure 5—figure supplement 2*). To test whether CCAP neurons are necessary for sugar and water ingestion, we reduced CCAP neurotransmission with nSyb RNAi, and measured ingestion of sugar or water. We found that silencing CCAP neurons decreased sugar ingestion and increased water ingestion (*Figure 5G*, *Figure 5—figure supplement 2*), demonstrating that CCAP neurons reciprocally regulate sugar and water ingestion, similar to the ISNs.

Although CCAP neurons are functionally connected to the ISNs, their synaptic connectivity is indirect. We identified the CCAP neurons in the FAFB volume (*Figure 5A*) and found weak connections between CCAP neurons and ISN synaptic partners: Cowboy (5 synapses), VESa1 (22 synapses), and a

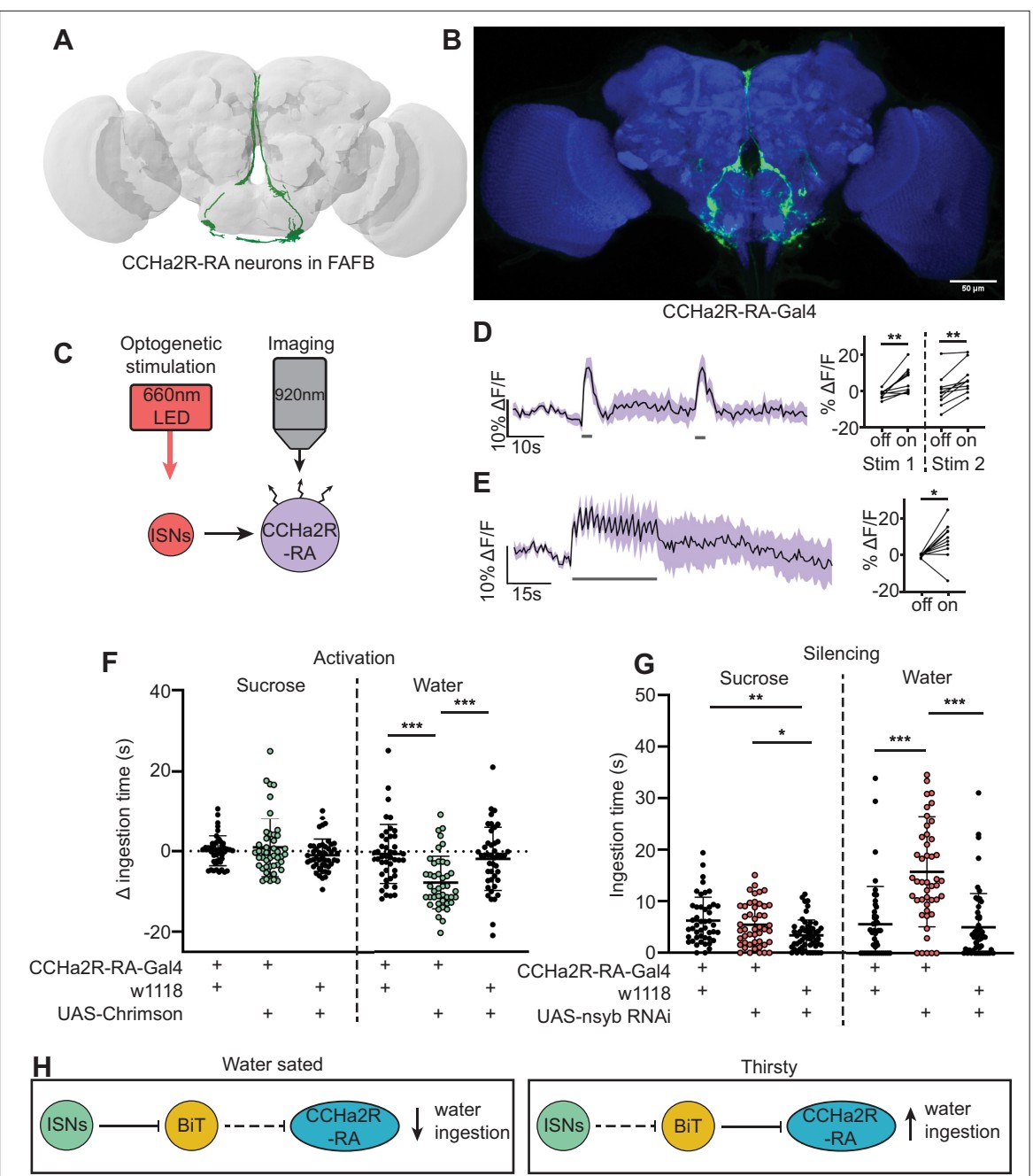

**Figure 4.** CCHamide-2 receptor isoform RA (CCHa2R-RA) neurons regulate water but not sugar ingestion and are likely inhibited by bilateral T-shaped neuron (BiT). (**A**) CCHa2R-RA neurons reconstruction from full adult fly brain (FAFB) dataset. (**B**) Light microscopy image of CCHa2R-RA-Gal4. (**C**) Experimental setup for in vivo calcium imaging. We expressed the light-sensitive ion channel Chrimson in the interoceptive subesophageal zone neurons (ISNs) and optogenetically stimulated them with 660 nm LED. We expressed the calcium sensor GCaMP in the CCHa2R-RA neurons and imaged them with a two-photon microscope. (**D**) Calcium responses of CCHa2R-RA neurites in subesophageal zone (SEZ) to 2 s optogenetic stimulation of the ISNs or (**E**) 30 s optogenetic stimulation of the ISNs. Left: Scatter plot shows mean ± SEM of all flies imaged, gray bars represent LED stimulation. Right: Quantification of mean fluorescence intensity before stim (off) and during stim (on), each dot represents one fly. Paired t-test and paired Wilcoxon test. n=10 flies. (**F**) Temporal consumption assay for 1 M sucrose or water during acute optogenetic activation of CCHa2R-RA neurons with Chrimson. Ingestion time of females exposed to light normalized to dark controls of indicated genotype. Sucrose: Kruskal-Wallis with Dunn's multiple comparison test. Water: One-way ANOVA with Holm-Šídák multiple comparison test. n=42–47 animals/genotype. (**G**) Temporal consumption assay for 1 M sucrose or water using RNAi targeting nSynaptobrevin (nSyb) in CCHa2R-RA neurons. Kruskal-Wallis with Dunn's multiple comparison test. n=45–54 animals/genotype. (**H**) Neural model for CCHa2R-RA regulation of water intake. Dashed lines indicate inactive synapses. *p<0.05, **p<0.01, ***p<0.001.

The online version of this article includes the following source data and figure supplement(s) for figure 4:

*Figure 4 continued on next page*

*Figure 4 continued*

**Source data 1.** CCha2R-RA functional imaging.

**Figure supplement 1.** CCHamide-2 receptor isoform RA (CCHa2R-RA) genetic controls functional imaging and response to bilateral T-shaped neuron (BiT) optogenetic stimulation.

**Figure supplement 2.** CCHamide-2 receptor isoform RA (CCHa2R-RA) optogenetic activation ingestion phenotype.

novel neuron we named BiT 2, based on its anatomical similarities to BiT (37 synapses). In addition, the ISN third-order neuron CCHa2R-RA neurons provide 26 synapses onto CCAP neurons (*Supplementary file 3*). This connectivity suggests that CCAP neurons are part of the broad network that receives ISN input (*Figure 5H*). Moreover, the reciprocal regulation of sugar and water ingestion by CCAP neurons argues that multiple peptidergic neurons downstream of the ISNs cooperate to coordinate ingestion of sugar versus water based on specific need.

## Discussion

In this study, we report that the ISNs communicate hunger and thirst states to a complex neural network that reaches several brain regions to regulate sugar and water ingestion (*Figure 6*). The ISNs synapse with neurons that project to higher brain neuroendocrine centers, including BiT, a novel neuron that reciprocally regulates sugar and water ingestion. Several peptide-releasing and peptide-sensing neurons known to regulate feeding behavior also receive ISN signals, providing the capacity to integrate hunger and thirst signals with many internal signals of nutritional need. These peptidergic neurons, connected to the ISNs via interneurons, contribute differentially to ingestion of sugar and water, with IPC and CCAP neurons reciprocally regulating sugar and water ingestion and CCHa2R-RA neurons modulating water ingestion. Thus, our work argues that the coordinated regulation of a peptidergic network weighs nutrient needs to generate nutrient-specific ingestion.

### The ISNs influence activity of several brain regions involved in feeding and nutrient homeostasis to coordinate sugar and water ingestion

Previous studies showed that the ISNs sense the hunger signal AKH and changes in hemolymph osmolality associated with thirst to correspondingly alter ISN neural activity. Increased ISN activity promotes sugar ingestion and decreases water ingestion, and decreased ISN activity decreases sugar ingestion and increases water ingestion (*Jourjine et al., 2016*). Here, we investigated how ISN activity reciprocally regulates sugar and water ingestion according to internal needs by examining the neural network modulated by the ISNs.

We found that the ISNs are predicted to synapse with 100 neurons, including projection neurons that arborize in neuroendocrine centers, SEZ interneurons, and ascending and descending neurons that likely innervate the ventral nerve cord. The majority of the ISN predicted synaptic partners are projection neurons that send arbors via the median bundle to the SMP, a neuroendocrine center (*Hartenstein, 2006*). This includes the cell-type BiT characterized in this study that reciprocally regulates sugar and water ingestion. Local SEZ neurons downstream of the ISNs include DSOG1, which are GABAergic and inhibit consumption (*Pool et al., 2014*), consistent with the notion that ISN activity directly influences feeding motor programs. In addition, eight uncharacterized descending neurons are downstream of the ISNs, suggesting that they may coordinate feeding with other motor behaviors, such as locomotion or digestion. While the number of ISN postsynaptic partners precludes comprehensive functional and behavioral analysis, the restricted number of brain regions that are direct targets of the ISNs (SMP, SEZ, and possibly ventral nerve cord) is consistent with ISN activity directly regulating neuroendocrine centers and feeding behavior.

We characterized the pathway from the second-order BiT projection neuron that oppositely regulates sugar and water consumption. We found that BiT has 93 predicted synaptic partners, including IPCs which are known to modulate food intake (*Nässel and Zandawala, 2020*), FLAa3/Lgr3 which have been implicated in regulating ingestion (*Laturney et al., 2022*; *Meissner et al., 2016*; *Nässel et al., 2013*; *Yeom et al., 2021*), and neurons labeled by the CCHa2R-RA-Gal4 (*Deng et al., 2019*) which we found to regulate water ingestion. BiT downstream neurons innervate several neuropils including the SEZ, SMP, SLP, FB, and lobula. Therefore, hunger and thirst signals sensed by the ISNs

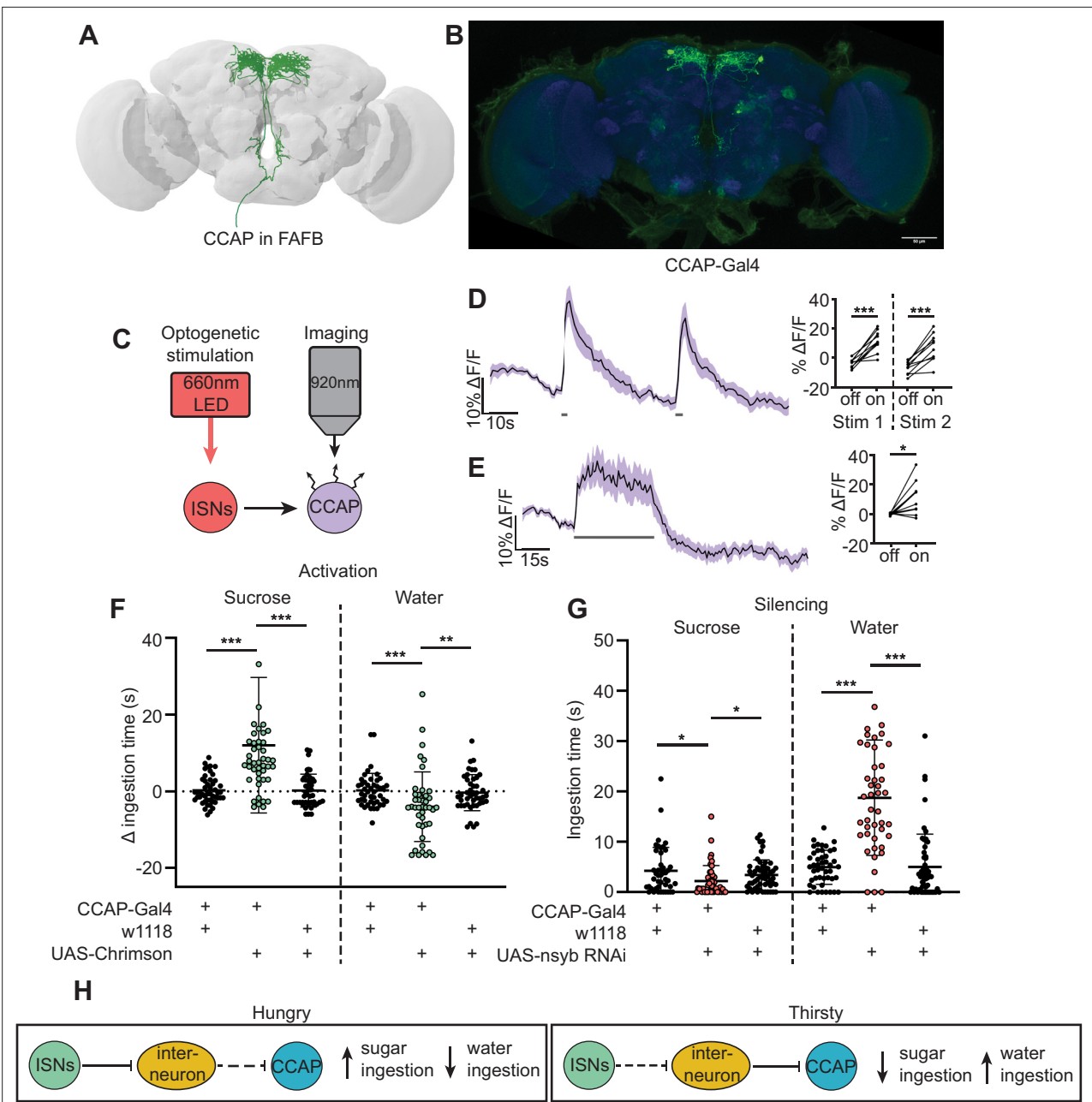

**Figure 5.** Crustacean cardioactive peptide (CCAP) neurons are downstream of the interoceptive subesophageal zone neurons (ISNs) and oppositely regulate sugar and water ingestion. (**A**) CCAP neurons reconstruction from full adult fly brain (FAFB) dataset. (**B**) Light microscopy image of CCAP-Gal4. (**C**) Experimental setup for in vivo calcium imaging. We expressed the light-sensitive ion channel Chrimson in the ISNs and optogenetically stimulated them with 660 nm LED. We expressed the calcium sensor GCaMP in the CCAP neurons and imaged them with a two-photon microscope. (**D**) Calcium response of CCAP neurites to 2 s optogenetic stimulation of the ISNs or (**E**) 30 s optogenetic stimulation of the ISNs. Left: Scatter plot shows mean ± SEM of all flies imaged, gray bars represent LED stimulation. Right: Quantification of mean fluorescence intensity before stim (off) and during stim (on), each dot represents one fly. Paired t-test. n=10 flies. (**F**) Temporal consumption assay for 1 M sucrose or water during acute optogenetic activation of CCAP neurons with Chrimson. Ingestion time of females exposed to light normalized to dark controls of indicated genotype. Sucrose: Kruskal-Wallis with Dunn's multiple comparison test, Water: One-way ANOVA with Holm-Šídák multiple comparison test. n=42–48 animals/genotype. (**G**) Temporal consumption assay for 1 M sucrose or water using RNAi targeting nSynaptobrevin (nSyb) in CCAP neurons. Kruskal-Wallis with Dunn's multiple comparison test. n=45–54 animals/genotype. (**H**) Neural model for CCAP coordination of sugar and water intake. Dashed lines indicate inactive synapses. *p<0.05, **p<0.01, ***p<0.001.

The online version of this article includes the following source data and figure supplement(s) for figure 5:

**Source data 1.** CCAP functional imaging.

*Figure 5 continued on next page*

*Figure 5 continued*

**Figure supplement 1.** Crustacean cardioactive peptide (CCAP) responds to interoceptive subesophageal zone neuron (ISN) activation using another CCAP-Gal4 driver.

**Figure supplement 2.** Crustacean cardioactive peptide (CCAP) optogenetic activation ingestion phenotype.

fan out to modulate multiple brain regions via BiT. We speculate that the broad reach of the ISNs serves to modulate different behaviors such as sleep, reproduction, and locomotion based on the hunger or thirst state of the fly.

## Communication between peptidergic neurons coordinates ingestion

Our studies demonstrate that multiple peptidergic neurons participate in regulation of sugar and water ingestion. We find that dILP3 RNAi or *amontillado* RNAi expression in the ISNs recapitulates the ISN loss-of-function phenotype, arguing that the ISNs themselves are peptidergic and utilize dILP3 as the neurotransmitter that conveys hunger and thirst signals. The ISNs have increased activity upon AKH detection or low osmolality (hunger signals) (*Jourjine et al., 2016*), arguing that increased dILP3 release from the ISNs drives sucrose ingestion and limits water ingestion in hungry flies to maintain homeostasis. This conversion of an AKH signal to a dILP3 signal resembles findings in *Drosophila* larvae, where circulating AKH binds to the AKH receptor on IPCs to release dILP3 and promote sucrose consumption (*Kim and Neufeld, 2015*; *Palovcik et al., 1984*).

The ISNs modulate activity in many neuroendocrine cells, potentially causing widespread changes in peptide release (*Nässel and Zandawala, 2022*; *Schlegel et al., 2016*). We find that ISN activation increases activity of CCAP neurons and CCHa2R-RA neurons, and BiT activation decreases the activity of IPCs. CCAP neurons are orexigenic and communicate to CCAP receptor cells, including IPCs (*Zhang et al., 2022b*) and a subpopulation of neuropeptide F (NPF) neurons (*Williams et al., 2020*). While this is the first study that characterizes the CCHa2R-RA neurons, the knockin Gal4 line that labels the CCHa2R-RA neurons was generated for the RA isoform of CCHa2 receptor, suggesting that these neurons respond to CCHa2, a peptide produced in the midgut and brain that increases appetite (*Deng et al., 2019*; *Ida et al., 2012*; *Reiher et al., 2011*; *Ren et al., 2015*). Therefore, CCHa2R-RA neurons potentially integrate the hunger and thirst signals from the ISNs with CCHa2 signals from the gut. IPCs are central regulators of appetite and metabolism, receive multiple direct and indirect signals of nutrient status, and release dILP2, dILP3, and dILP5 (*Nässel and Zandawala, 2020*). Our finding that the ISNs communicate with multiple peptidergic systems argues that hunger and thirst signals sensed by the ISNs are integrated with other nutritive state signals

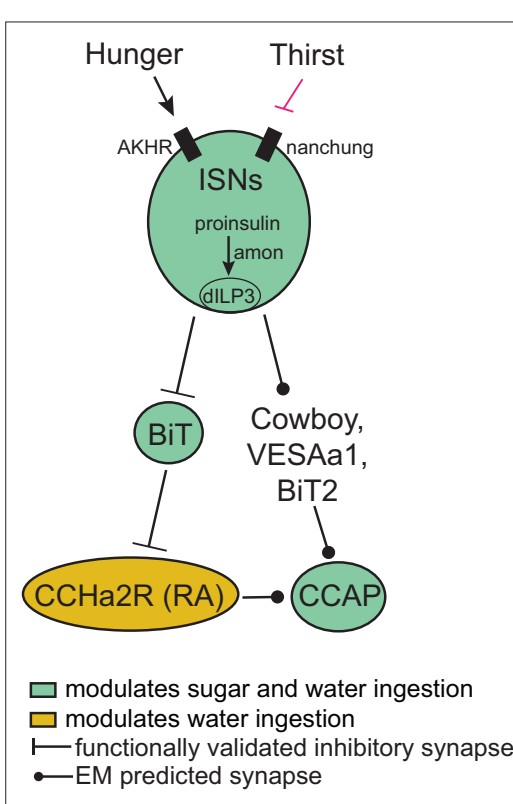

**Figure 6.** Interoceptive subesophageal zone neurons (ISN) regulation of sugar and water ingestion model. Hunger signals activate the ISN while thirst signals inhibit the ISNs. ISNs use *Drosophila* insulin-like peptide 3 (dILP3) as a neurotransmitter and require amontillado (amon) for neuropeptide processing. ISN activity inhibits bilateral T-shaped neuron (BiT), which in turn inhibits CCHamide-2 receptor isoform RA (CCHa2R-RA) neurons. Crustacean cardioactive peptide (CCAP) neurons are downstream of the ISNs, connected via Cowboy, VESAa1, BiT2, and CCHa2R-RA neurons. BiT activity inhibits sugar ingestion and promotes water ingestion. CCAP activity promotes sugar ingestion and inhibits water ingestion. CCHa2R-RA activity inhibits water ingestion.

sensed by ISN downstream neurons for a global assessment of the current nutritional demands of the animal.

## Sugar and water ingestion remains coordinated downstream of the ISNs

Multiple neurons downstream of the ISNs bidirectionally regulate both sugar and water ingestion, arguing that they bias ingestion based on nutrient need. By studying the activation and silencing phenotypes associated with CCAP neurons, we show that acute activation promotes sugar ingestion and limits water ingestion, while silencing these neurons has the opposite effects. These findings are consistent with and expand upon previous studies showing that CCAP neurons promote feeding (*Selcho et al., 2018*; *Williams et al., 2020*). IPCs have a more complex role in regulating ingestion, with several studies showing that their acute activation limits ingestion of sucrose or food (*Nässel et al., 2015*; *Semaniuk et al., 2018*; *Wang et al., 2020*) and other studies suggesting the opposite (*Sudhakar et al., 2020*). We find that under the specific conditions of our assay, acute activation of IPCs promotes sucrose ingestion and limits water ingestion. We suspect that differing findings upon IPC manipulation may stem from differences in the deprivation state of the fly, the behavioral assay, the type and timing of neural manipulation, and the food source. As IPCs receive multiple internal state signals, it is possible that activation phenotypes depend on the current state of IPC modulation set by the internal state of the fly.

Overall, our work sheds light on neural circuit mechanisms that translate internal nutrient abundance cues into the coordinated regulation of sugar and water ingestion. We show that the hunger and thirst signals detected by the ISNs influence a network of peptidergic neurons that act in concert to prioritize ingestion of specific nutrients based on internal needs. We hypothesize that multiple internal state signals are integrated in higher brain regions such that combinations of peptides and their actions signify specific needs to drive ingestion of appropriate nutrients. As peptide signals may act at a distance and may cause long-lasting neural activity state changes, studying their integration over space and time is a future challenge to further illuminate homeostatic feeding regulation.

## Materials and methods

**Key resources table**

| Reagent type (species) or resource | Designation | Source or reference | Identifiers | Additional information |
|---|---|---|---|---|
| Genetic reagent (*D. melanogaster*) | UAS-nSynaptobrevin RNAi | Bloomington Drosophila Stock Center | BDSC 31983 | |
| Genetic reagent (*D. melanogaster*) | UAS-dcr2 | Bloomington Drosophila Stock Center | BDSC 24650 | |
| Genetic reagent (*D. melanogaster*) | UAS-Trh RNAi | Bloomington Drosophila Stock Center | BDSC 25842 | |
| Genetic reagent (*D. melanogaster*) | UAS-ChAT RNAi | Bloomington Drosophila Stock Center | BDSC 25856 | |
| Genetic reagent (*D. melanogaster*) | UAS-Tbh RNAi | Bloomington Drosophila Stock Center | BDSC 27667 | |
| Genetic reagent (*D. melanogaster*) | UAS-Hdc RNAi | Bloomington Drosophila Stock Center | BDSC 26000 | |
| Genetic reagent (*D. melanogaster*) | UAS-VMAT RNAi | Bloomington Drosophila Stock Center | BDSC 31257 | |
| Genetic reagent (*D. melanogaster*) | UAS-GAD1 RNAi | Bloomington Drosophila Stock Center | BDSC 28079 | |
| Genetic reagent (*D. melanogaster*) | UAS-DDC RNAi | Bloomington Drosophila Stock Center | BDSC 27030 | |
| Genetic reagent (*D. melanogaster*) | UAS-DVGlut RNAi | Bloomington Drosophila Stock Center | BDSC 27538 | |

*Continued on next page*

*Continued*

| Reagent type (species) or resource | Designation | Source or reference | Identifiers | Additional information |
|---|---|---|---|---|
| Genetic reagent (*D. melanogaster*) | UAS-sNPF RNAi | Bloomington Drosophila Stock Center | BDSC 25867 | |
| Genetic reagent (*D. melanogaster*) | UAS-VGAT RNAi | Bloomington Drosophila Stock Center | BDSC 41958 | |
| Genetic reagent (*D. melanogaster*) | UAS-TDC2 RNAi | Bloomington Drosophila Stock Center | BDSC 25871 | |
| Genetic reagent (*D. melanogaster*) | UAS-dILP1 RNAi | Bloomington Drosophila Stock Center | BDSC 32861 | |
| Genetic reagent (*D. melanogaster*) | UAS-dILP2 RNAi | Bloomington Drosophila Stock Center | BSC 32475 | |
| Genetic reagent (*D. melanogaster*) | UAS-dILP3 RNAi | Bloomington Drosophila Stock Center | BSC 31492 | |
| Genetic reagent (*D. melanogaster*) | UAS-dILP4 RNAi | Bloomington Drosophila Stock Center | BDSC 33682 | |
| Genetic reagent (*D. melanogaster*) | UAS-dILP5 RNAi | Bloomington Drosophila Stock Center | BDSC 31378 | |
| Genetic reagent (*D. melanogaster*) | UAS-dILP6 RNAi | Bloomington Drosophila Stock Center | BDSC 33684 | |
| Genetic reagent (*D. melanogaster*) | UAS-dILP7 RNAi | Bloomington Drosophila Stock Center | BDSC 32862 | |
| Genetic reagent (*D. melanogaster*) | UAS-amon RNAi | Bloomington Drosophila Stock Center | BDSC 29009 | |
| Genetic reagent (*D. melanogaster*) | ISN-Gal4 (VT011155-Gal4) | FlyLight, Janelia Research Campus | Fly Light ID 54404 | |
| Genetic reagent (*D. melanogaster*) | ISN-LexA (GMR34G02-LexA) | Bloomington Drosophila Stock Center | BDSC 54138 | |
| Genetic reagent (*D. melanogaster*) | UAS-myrGFP.QUAS-mtdTomato-3xHA; trans-Tango | Bloomington Drosophila Stock Center | BDSC 77124 | |
| Genetic reagent (*D. melanogaster*) | VT002073-Gal4.AD | Bloomington Drosophila Stock Center | BDSC 71871 | |
| Genetic reagent (*D. melanogaster*) | VT040568-Gal4.DBD | Bloomington Drosophila Stock Center | BDSC 72902 | |
| Genetic reagent (*D. melanogaster*) | UAS-csChrimson.mVenus | Bloomington Drosophila Stock Center | BDSC 55134 | |
| Genetic reagent (*D. melanogaster*) | LexAop-ChrimsonR.mCherry | Gift from Jayaraman Lab | | |
| Genetic reagent (*D. melanogaster*) | UAS-ArcLight | Bloomington Drosophila Stock Center | BDSC 51056 | |
| Genetic reagent (*D. melanogaster*) | Empty split | Bloomington Drosophila Stock Center | BDSC 79603 | |
| Genetic reagent (*D. melanogaster*) | ppk28-Gal4 | *Cameron et al., 2010*. | BDSC 93020 | |
| Genetic reagent (*D. melanogaster*) | Gr5a-Gal4 | *Chyb et al., 2003*. | BDSC 57592, 57591 | |
| Genetic reagent (*D. melanogaster*) | CCha2R-RA-Gal4 | Bloomington Drosophila Stock Center | BDSC 84603 | |
| Genetic reagent (*D. melanogaster*) | LexAop-CsChrimson.tdTomato (III) | Bloomington Drosophila Stock Center | BDSC 82183 | |

*Continued on next page*

*Continued*

| Reagent type (species) or resource | Designation | Source or reference | Identifiers | Additional information |
|---|---|---|---|---|
| Genetic reagent (*D. melanogaster*) | UAS-GCaMP6s (III) | Bloomington Drosophila Stock Center | BDSC 42749 | |
| Genetic reagent (*D. melanogaster*) | 20XUAS-GCaMP7b | Bloomington Drosophila Stock Center | BDSC 79029 | |
| Genetic reagent (*D. melanogaster*) | CCAP-Gal4 (II) | Bloomington Drosophila Stock Center | BDSC 25685 | |
| Genetic reagent (*D. melanogaster*) | CCAP-Gal4 (III) | Bloomington Drosophila Stock Center | BDSC 25686 | |
| Genetic reagent (*D. melanogaster*) | CCHa2R (RA)-LexA | Bloomington Drosophila Stock Center | BDSC 84363 | |
| Genetic reagent (*D. melanogaster*) | dILP2-LexA | *Li and Gong, 2015*. | | |
| Antibody | Anti-Brp (nc82) (Mouse monoclonal) | DSHB, University of Iowa, USA | DSHB Cat# nc82, RRID:AB_2314866 | 1:40 |
| Antibody | Anti-GFP (Chicken polyclonal) | Invitrogen | Thermo Fisher Scientific Cat# A10262, RRID:AB_2534023 | 1:1000 |
| Antibody | Anti-dsRed (Rabbit polyclonal) | Takara, Living Colors | Takara Bio Cat# 632496, RRID:AB_10013483 | 1:1000 |
| Antibody | Anti-mouse AF647 (Goat polyclonal) | Invitrogen | Thermo Fisher Scientific Cat# A-21236, RRID:AB_2535805 | 1:500 |
| Antibody | Anti-chicken AF488 (Goat polyclonal) | Life Technologies | Thermo Fisher Scientific Cat# A-11039, RRID:AB_2534096 | 1:1000 |
| Antibody | Anti-rabbit AF568 (Goat polyclonal) | Invitrogen | Thermo Fisher Scientific Cat# A-11036, RRID:AB_10563566 | 1:1000 |
| Chemical compound | All trans-Retinal | MilliporeSigma | Cat# R2500 | |
| Software, algorithm | Fiji | https://fiji.sc/ | RRID: SCR_002285 | |
| Software, algorithm | NAVis | Copyright 2018, Philipp Schlegel | | |
| Software, algorithm | CATMAID | *Saalfeld et al., 2009*; https://catmaid.org | | |
| Software, algorithm | Flywire | Flywire; https://flywire.ai/ | RRID:SCR_019205 | |
| Software, algorithm | GraphPad Prism | GraphPad Software; https://www.graphpad.com/scientific-software/prism/ | RRID:SCR_002798 | |
| Software, algorithm | Python | Python Software Foundation; https://www.python.org/downloads/ | | |
| Software, algorithm | Adobe Illustrator | Adobe Software; https://www.adobe.com/products/illustrator.html | | |
| Software, algorithm | CAVE (connectome annotation versioning engine) | https://github.com/seung-lab/CAVEclient/blob/master/FlyWireSynapseTutorial.ipynb; *seung-lab, 2021*; *Buhmann et al., 2021*; *Eckstein et al., 2020*; *Heinrich et al., 2018* | | |
| Software, algorithm | R Project for Statistical Computing | *Dessau and Pipper, 2008* | RRID:SCR_001905 | |

## Fly husbandry

All experiments and screening were carried out with adult *D. melanogaster* females reared on standard cornmeal-agar-molasses medium, at 25°C, 65–70% humidity, on a 12 hr light:12 hr dark cycle. Flies used in optogenetic assays were reared on food containing 0.25 mM all-trans-retinal (Sigma-Aldrich) in darkness, before and after eclosion. See *Supplementary file 4* for all fly genotypes.

## Temporal consumption assay

Flies were anesthetized using $CO_2$ and then fixed to a glass slide with nail polish. Flies recovered for 2 hr in a humidified box if testing for sucrose ingestion, or in a desiccated box with Drierite if testing

for water ingestion. Immediately before testing for sucrose ingestion, flies were given water until they no longer responded to three consecutive water presentations. In testing, flies were presented with the tastant (water or 1 M sucrose) 10 times and consumption time was manually recorded. All experiments were done in a dark, temperature- and humidity-controlled room. IR lights and IR cameras were used to conduct experiments in the dark. All water tests were done in the morning and all sugar tests were done in the afternoon. For optogenetic activation experiments, we expressed the light activated ion channel Chrimson in the neurons of interest and activated these neurons using a 635 nm laser (Laserglow). For silencing experiments, we expressed RNAi against nSyb in neurons of interest. All experiments were performed blind to the genotype being tested and across 3–4 days. Flies were excluded from analysis if during the experiment they were covered in sugar or water droplet.

## In vivo calcium imaging

Calcium imaging studies were carried out as described in *Shiu et al., 2022*. Mated female flies were dissected for calcium imaging studies 5–14 days post-eclosion. Flies were briefly anesthetized with ice and placed in a custom plastic holder at the neck to isolate the head from the rest of the body. The head was then immobilized using UV glue, the proboscis was immobilized using wax, and the esophagus was cut to provide unobstructed imaging access to the SEZ. All flies imaged were sated. In vivo calcium imaging with optogenetic activation was performed in a two-photon microscope using a Scientifica Hyperscope with resonant scanning, a piezo drive, and a 20× water immersion objective (NA = 1.0) with 1.8–3× digital zoom, depending on the cell type imaged. Calcium responses were recorded with a 920 nm laser and optogenetic stimulation was achieved with a 660 nm LED. Two s LED stimulation paradigm: 20 s off, 2 s on, 30 s off, 2 s on, 30 s off. 30 s LED stimulation paradigm: 20 s off, (1 s on, 1 s off) × 15, 60 s off. For the 2 s LED stimulation, 80 stacks of 20 z slices of 4–5 μm were acquired at 0.667 Hz. For the 30 s stimulation, 125 stacks of 20 z slices of 4–5 μm were acquired at 0.667 Hz. Analysis was done on max-z projections of the 20 z slices. $\%\Delta F/F=100*((Ft - F0)/F0)$, where Ft is the fluorescence of the neuron ROI - the background ROI at each timepoint and F0 is the mean Ft for the 23 timepoints prior to stimulus onset. Quantification was carried out in GraphPad Prism. A mean fluorescence intensity for LED off and LED on was calculated for each fly. For the 2 s LED stimulation, mean intensity for LED off was calculated for five timepoints immediately before LED exposure and mean intensity for LED on was calculated for five timepoints during LED exposure. For the 30 s stimulation, mean intensity for LED off was calculated for 28 timepoints immediately before LED exposure and mean intensity for LED on was calculated for 28 timepoints during LED exposure. Paired t-test or paired Wilcoxon test was performed. ROI for CCHa2R-RA imaging was CCHa2R-RA neurites in SEZ. ROI for CCAP imaging was CCAP neurites. ROI for IPC imaging was all IPC somas. All experiments per genotype were conducted across 2–4 days. Flies were excluded from analysis if excitotoxicity was detected.

## In vivo voltage imaging

Voltage imaging studies were performed for neurons predicted to be silenced by presynaptic neurons based on behavioral or EM connectivity data. Voltage imaging studies were carried out exactly as calcium imaging studies described above. ROI for BiT imaging was BiT soma.

## Immunohistochemistry

All brain and CNS dissections and immunostaining (unless directly addressed) were carried out as described (https://www.janelia.org/project-team/flylight/protocols, 'IHC-Anti-GFP') substituting the below antibodies and eschewing the pre-embedding fixation steps. Ethanol dehydration and DPX mounting was carried out as described (https://www.janelia.org/project-team/flylight/protocols, 'DPX Mounting').

Primary antibodies:

- mouse α-Brp (nc82, DSHB, University of Iowa, USA) at 1:40
- chicken α-GFP (Invitrogen, A10262) at 1:1000
- rabbit α-dsRed (Takara, Living Colors 632496) at 1:1000

Secondary antibodies:

- goat α-mouse AF647 (Invitrogen, A21236) at 1:500

- goat α-chicken AF488 (Life Technologies, A11039) at 1:1000
- goat α-rabbit AF568 (Invitrogen, A21236) at 1:1000

See Key resources table for additional information on antibodies. Images were acquired with a Zeiss LSM 880 NLO AxioExaminer with Airyscan and Coherent Chameleon Vision or Zeiss LSM 780 Laser Scanning Confocal Microscope at the Berkeley Molecular Imaging Center with a Plan-Apochromat 20×/1.0 W, 40× W, 40×/1.4 oil, or 63×/1.4 oil objective. Images were prepared in Fiji.

### Electron microscopy neural reconstructions and connectivity

Neurons were reconstructed in a serial section transmission electron volume (full adult female brain, Zheng and Lauritzen et al., 2018) using the CATMAID software (Saalfeld et al., 2009). Fully manual reconstructions were generated by following the branches of the neuron and marking the center of each branch, thereby creating a 'skeleton' of each neuron. In addition to fully manual reconstructions, segments of an automated segmentation (Li et al., 2019) were proofread and expanded to generate complete reconstructions. In addition to the skeleton tracing, new chemical synapses were also annotated as previously described (Zheng et al., 2018). Downstream synaptic targets of the ISNs and BiT were then traced out from these additional locations using both manual and assisted tracing techniques as described above. Neurons traced in CATMAID, including ISNs and BiT, were all located in Flywire (flywire.ai), which uses the same EM dataset (Zheng et al., 2018). To identify synaptic partners, we used connectome annotation versioning engine (CAVE, Buhmann et al., 2021; Ida et al., 2012) using a cleft score cutoff of 50 to generate synapses of relatively high confidence (Ida et al., 2012; Baker et al., 2022). FAFB neural reconstructions were visualized using NAVis (Copyright 2018, Philipp Schlegel), which is based on natverse (Bates et al., 2020). See Key resources table for additional software information.

### BiT split-Gal4 generation

We created a color depth max intensity projection (CDM) mask of BiT reconstructed EM skeleton and used CDM mask searching (Otsuna et al., 2018) to find enhancers whose expression patterns seemed to include the desired cell type using MCFO (Nern et al., 2015) screening of subsets of the Janelia Research Campus and Vienna Tile Gal4 collections. Construction of stable split-Gal4 lines was performed as previously described (Dionne et al., 2018; Sterne et al., 2021). Immunohistochemistry and confocal imaging was used to determine successful split-Gal4 combinations.

### Identification of GAL4 lines from EM reconstructions

Visual inspection of Gal4 collections was used to determine cell type. Images of potential Gal4 lines were skeletonized in FIJI, converted into .swc format using natverse (Bates et al., 2020), and uploaded to Flywire using the Flywire Gateway. This generated pointclouds that were used to identify the neurons of interest. As Flywire permits exhaustive searching of neurons in an area, we examined all neurons in the region of interest to conclusively identify our neuron of interest.

### Statistical analysis

Statistical tests were performed in GraphPad Prism. For all group comparisons, data was first tested for normality with the KS normality test (alpha = 0.05). If all groups passed then groups were compared with a parametric test, but if at least one group did not pass, groups were compared with a non-parametric version. All statistical tests, significance levels, and number of data points (N) are specified in the figure legend.

All datasets from optogenetic behavior assays were normalized within each genotype. To generate this normalized dataset, data from females within the no light condition was averaged, creating a 'no-light mean' for each genotype. This value was subtracted from each individual female within the light condition of the corresponding genotype. This dataset was then graphed, and statistical analyses were performed as outlined above.

## Acknowledgements

Members of the Scott lab provided contributions to experimental design, data analysis, and manuscript preparation. This work was supported by NIH R01GM128209 (KS), R01GM128209 Diversity

Supplement (AGS), UC LEADs fellowship (ADT), and International Fellowship for CONICET Researchers (GP). Neuronal reconstruction for this project took place in a collaborative CATMAID environment in which 27 labs are participating to build connectomes for specific circuits. Development and administration of the FAFB tracing environment and analysis tools were funded in part by National Institutes of Health BRAIN Initiative grant 1RF1MH120679-01 to Davi Bock and Greg Jefferis, with software development effort and administrative support provided by Tom Kazimiers (Kazmos GmbH) and Eric Perlman (Yikes LLC). We thank Peter Li, Viren Jain, and colleagues at Google Research for sharing the automatic segmentation (*Li et al., 2019*). Tracing in Cambridge was supported by Wellcome Trust (203261/Z/16/Z) and ERC (649111) awards to G Jefferis. Neurons were also reconstructed and proofread in FlyWire, where we also identified pre- and postsynaptic partners. We acknowledge the Princeton FlyWire team and members of the Murthy and Seung labs for development and maintenance of FlyWire (supported by BRAIN initiative grant MH117815 to Murthy and Seung). In addition to our tracing efforts in CATMAID and proofreading in FlyWire, the following labs greatly contributed to the proofreading in FlyWire of the neurons characterized in this study: Murthy and Seung Labs (76.06%), Jefferis Lab (13.57%) and Jinseop Kim Lab (7.21%). We are appreciative of the proofreading contributed by other labs including the Anderson, Bock, Dacks, Dickson, Huetteroth, Pankratz, Seeds/Hampel, Selcho, Simpson, Waddell, Wilson and Wolf Labs, and Janelia tracers (*Supplementary files 1–3*). Vivek Jayaraman provided unpublished fly lines used in this study. We thank Stefanie Engert for tracing two ISNs in CATMAID, Zepeng Yao for identifying CCHa2R-RA neurons, Phil Shiu for identifying DSOG1 neurons, and Amanda Abusaif for her tracing and proofreading efforts. Confocal imaging experiments were conducted at the CRL Molecular Imaging Center, RRID:SCR_017852, supported by NSF DBI-1041078 and the Helen Wills Neuroscience Institute. We thank Holly Aaron and Feather Ives for microscopy training and support.

## Additional information

### Funding

| Funder | Grant reference number | Author |
| --- | --- | --- |
| National Institutes of Health | R01GM128209 | Kristin Scott |
| National Institutes of Health | R01GM128209 Diversity Supplement | Amanda J González Segarra |
| Consejo Nacional de Investigaciones Científicas y Técnicas | International Fellowship | Gina Pontes |
| University of California Berkeley | LEADS fellowship | Alexander Del Toro |

The funders had no role in study design, data collection and interpretation, or the decision to submit the work for publication.

### Author contributions

Amanda J González Segarra, Conceptualization, Data curation, Formal analysis, Supervision, Funding acquisition, Validation, Investigation, Visualization, Methodology, Writing – original draft, Writing – review and editing; Gina Pontes, Data curation, Formal analysis, Funding acquisition, Investigation, Visualization; Nicholas Jourjine, Investigation, Visualization; Alexander Del Toro, Funding acquisition, Investigation, Visualization; Kristin Scott, Conceptualization, Supervision, Funding acquisition, Methodology, Writing – original draft, Project administration, Writing – review and editing

### Author ORCIDs
Kristin Scott https://orcid.org/0000-0003-3150-7210

Reviewer #1 (Public Review): https://doi.org/10.7554/eLife.88143.3.sa1
Reviewer #2 (Public Review): https://doi.org/10.7554/eLife.88143.3.sa2

Author Response https://doi.org/10.7554/eLife.88143.3.sa3

## Additional files

### Supplementary files

- MDAR checklist

- Supplementary file 1. Interoceptive subesophageal zone neuron (ISN) postsynaptic partners. Number of synapses from the four ISNs onto different cell types, including the Flywire tracing contributions of different laboratories.

- Supplementary file 2. Bilateral T-shaped neuron (BiT) postsynaptic neurons. Number of synapses from BiT onto different cell types, including the Flywire tracing contributions of different laboratories.

- Supplementary file 3. Synapses from interoceptive subesophageal zone neuron (ISN) postsynaptic partners onto crustacean cardioactive peptide (CCAP) neurons. Number of synapses from ISN postsynaptic partners onto CCAP neurons, including the Flywire tracing contributions of different laboratories.

- Supplementary file 4. Fly genotypes used. Fly genotypes used throughout the study, as shown in different figure panels.

### Data availability

All data is included in the manuscript and supporting files. FAFB volume neurons can be accessed at flywire.ai. IDs for all flywire neurons are included in Source Data files.

The following previously published dataset was used:

| Author(s) | Year | Dataset title | Dataset URL | Database and Identifier |
|---|---|---|---|---|
| Zheng Z, Lauritzen JS, Perlman E, Robinson CG, Nichols M, Milkie D, Torrens O, Price J, Fisher CB, Sharifi N, Calle-Schuler SA, Kmecova L, Ali IJ, Karsh B, Trautman ET, Bogovic JA, Hanslovsky P, Kazhdan M, Khairy K, Saalfeld S, Fetter RD, Bock DD, Jefferies GS | 2018 | A Complete Electron Microscopy Volume of the Brain of Adult *Drosophila melanogaster* | https://catmaid-fafb.virtualflybrain.org/ | CATMAID, virtualflybrain |

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
