## [Editor Report · eLife assessment]

This **important** study identifies and characterizes a broad peptidergic network that coordinates nutrient-specific consumption needs for food or water. Using state-of-the-art methodology the authors combine a well-balanced set of exploratory anatomical analyses with rigorous functional experimental approaches to examine how ingestion is regulated based on internal needs. These significant and **convincing** new findings are of broad interest to the neuroscience field.

---

## [Referee Report · Reviewer #1 (Public Review)]

This work by Gonzalez-Segarra et al. greatly extends previous research from the same group that identified ISNs as a key player in balancing nutrition and water ingestion. Using well-balanced sets of exploratory anatomical analyses and rigorous functional experiments, the authors identify and compile various peptidergic circuits that modulate nutrient and/or water ingestion. The findings are convincing and the experiments rigorous.

Strengths:

- The authors complement anatomically-reconstructed and functionally-validated neuronal connectivity with extensive and intensive morphological and synaptic reconstruction.

- Neurons and genes involved in specific components of feeding control are undoubtedly challenging, because numerous neurons and circuits redundantly and reciprocally regulate the same components of feeding behavior. This work dissociates how multiple, parallel and interconnected, peptidergic circuits (dilp3, CCHa2, CCAP) modulate sucrose and water ingestion, in tandem and in parallel.

- The authors address some of the incongruencies/discrepancies in current literature (IPCs) and try to provide explanations, rather than ignoring inconsistent findings.

Weaknesses:

- The authors have addressed several weaknesses of the paper in the revised text.

---

## [Referee Report · Reviewer #2 (Public Review)]

In this manuscript, González-Segarra et al. investigated how ISNs regulate sugar and water ingestion in *Drosophila*. In their previous paper, authors have shown that inhibiting neurotransmission in ISNs has opposite effects on sugar and water ingestion. In this manuscript, the authors first identified the effector molecules released by ISNs. Their RNAi screen found that, surprisingly, ISNs use ilp3 as a neuromodulator. Next, using light and electron microscopy, they investigated the downstream neural circuits ISNs connect with to regulate water or sugar ingestion. These analyses identified a new group of neurons named Bilateral T-shaped neurons (BiT) as the main output of ISNs, and several other peptidergic neurons as downstream effectors of ISNs. While BiT activity regulated both sugar and water ingestion, BiT downstream neurons, such as CCHa2R, only impacted water ingestion. These results suggested that ISNs might interact with distinct neural circuits to control sugar or water ingestion. The authors also investigated other ISN downstream neurons, such as ilp2 and CCAP, and revealed that their activity also contributes to ingestive behaviors in flies.

Major strengths:

1. This manuscript presents a comprehensive investigation of the downstream neurons connected to ISNs.

2. The authors have identified and characterized a diverse set of peptidergic neurons that regulate ingestive behaviors in the fly brain.

Weaknesses:

1. Only one RNAi hairpin is used to knock down Ilp3 in ISNs? There is a concern about off-targeting effects without the presence of another hairpin or mutant data. Do ilp3 mutants also have similar defects in sugar/water ingestion compared to ISN ilp3 knockdown?

2. Throughout the paper, authors use either voltage or calcium sensors without explaining why they choose to use either method to determine the functional connectivity between neurons.

3. How these diverse sets of peptidergic neurons interact to regulate ingestive behaviors is unclear and requires further investigation.

---

## [Author Response]

The following is the authors’ response to the original reviews.

We greatly appreciate the positive feedback of the reviewers and have modified the manuscript to address their comments, including changes to the text, figures, and methods. We believe that these revisions have strengthened and improved the manuscript. Reviewers’ comments in blue and detailed responses in black are below.

**Reviewer #1 Weaknesses:**
Is "function" of the ISNs to balance "nutrient need" or osmolarity? Balancing hemolymph osmolarity for physiological homeostasis is conceptually different from balancing thirst and hunger.

We have added the following text to the introduction to address this: “Thus, the ISNs sense both AKH and hemolymph osmolality, arguing that they balance internal osmolality fluctuations and nutrient need (Jourjine, Mullaney et al., 2016).” (ln 80-82).

The final schematic nicely sums up how the different peptidergic pathways might work together, but it is unclear which connections are empirically-validated or speculative. It would be informative to show which parts of the model are speculative versus validated. For example, does FAFB volume synapse = functional connectivity and not just anatomical proximity? A bulk of the current manuscript relies on "synapses of relatively high confidence" (according to Materials and methods: line 522). I recommend distinguishing empirically tested & predicted connections in the final schematic, and maybe reword/clarify throughout the manuscript as "predicted synaptic partners"

We modified the schematic to clarify EM based connections versus functionally validated connections. We also clarified the EM predicted synaptic partners, using “predicted synaptic partners” throughout the manuscript.

**Reviewer #2 Areas for further development:**
• Does BIT inhibit all of the IPCs or some of them? I think it is critical to indicate the ROIs used for each neuron in the methods. Which part of the neuron is used for imaging experiments? Dendrites, cell bodies, or synaptic terminals?

ROIs used for quantification are described in the figure legends: “ArcLight response ofBiT soma…” (Fig 2, Fig S2), “Calcium responses of CCHa2R-RA neurites in SEZ…” (Fig 4), “Calcium response of CCHa2R-RA SEZ neurites…” (Fig S4), “Calcium response of CCAP neurites…” (Fig 5, Fig S5), “Calcium response of all IPC somas…” (Fig S3). We have added ROIs used for quantification to the ‘In vivo calcium imaging’ and the ‘In vivo voltage imaging’ methods sections (ln 493-494).

• The discussion section is not giving big picture explanation of how these neurons work together to regulate sugar and water ingestion. Silencing and activation experiments are good, but without showing the innate activity of these neural groups during ingestion, it is not clear what their functions are in terms of regulating fly behavior.

We agree that how these peptidergic neurons coordinately regulate feeding is unclear. As peptide signals may act at a distance and may cause long-lasting neural activity state changes, studying their integration over space and time is challenging. Acute imaging during feeding would only in part address this challenge, as cumulative changes in nutrient need signals may impart circuit changes that are not apparent by monitoring the acute activity of peptidergic neurons. We modified a paragraph in the discussion to address this (ln 434-443).

“Overall, our work sheds light on neural circuit mechanisms that translate internal nutrient abundance cues into the coordinated regulation of sugar and water ingestion. We show that the hunger and thirst signals detected by the ISNs influence a network of peptidergic neurons that act in concert to prioritize ingestion of specific nutrients based on internal needs. We hypothesize that multiple internal state signals are integrated in higher brain regions such that combinations of peptides and their actions signify specific needs to drive ingestion of appropriate nutrients. As peptide signals may act at a distance and may cause long-lasting neural activity state changes, studying their integration over space and time is a future challenge to further illuminate homeostatic feeding regulation.”

**Reviewer #1 (Recommendations For The Authors):**
For the final schematic figure, it may be informative to include nanchung and AKHR in the schematic.

We now include this (Fig 6).

For the ingestion duration with optogenetic activation, I don't think the right way to represent the data is by normalizing them to the no LED control. I think it should show raw ingestion time. I understand that the normalized data make the figure "cleaner" (no need to show +/- LED separately) but I think visualization of the raw data is important.

We now include this in a new Supplemental Figure (Fig S6).

Methods for ingestion with optogenetic activation should be detailed in the Methods section.

We expanded upon this in the ‘Temporal consumption assay (TCA)’ methods section. (ln 461-466).

**Reviewer #2 (Recommendations For The Authors):**
1. I think the authors are not following the recommendations of the Flywire community which recommends that people who contributed to the tracing of neurons are offered authorship in the published papers. I see the authors are thanking other lab members who have done tracing for the neurons described in this study, but I would like them to clarify whether they are following the guidelines provided by Flywire.

We followed the Flywire guidelines and contacted all Flywire users contributing more that 10% to neuron edits for permission to publish with acknowledgements. (see Flywire guidelines https://docs.google.com/document/d/1bUkOB5JnT3u__JDvAoVDHJ3zr5NXQtV_63yx2w6Tcc/edit).

1. The method section for voltage imaging is missing.

We now include a section on voltage imaging (ln 496-498).

1. ROIs for imaging are not indicated in the methods or in the figures. It is hard to judge what is the origin of neural activity plotted in the figures; are they imaging cell bodies, dendrites, or axons?

ROIs used for quantification are described in the figure legends: “ArcLight response ofBiT soma…” (Fig 2, Fig S2), “Calcium responses of CCHa2R-RA neurites in SEZ…” (Fig 4), “Calcium response of CCHa2R-RA SEZ neurites…” (Fig S4), “Calcium response of CCAP neurites…” (Fig 5, Fig S5), “Calcium response of all IPC somas…” (Fig S3). We have added ROIs used for quantification to the ‘In vivo calcium imaging’ and the ‘In vivo voltage imaging’ methods sections (ln 493-494).